# Vitamin B12 Status in Recreational Users of Nitrous Oxide: A Systematic Review Focusing on the Prevalence of Laboratory Abnormalities

**DOI:** 10.3390/antiox12061191

**Published:** 2023-05-31

**Authors:** Tanguy Ménétrier, Damien Denimal

**Affiliations:** 1Department of Biochemistry, University Hospital of Dijon, F-21079 Dijon, France; tanguy_menetrier@etu.u-bourgogne.fr; 2UMR1231 LNC INSERM, University of Burgundy, F-21079 Dijon, France

**Keywords:** cobalamins, biomarkers, oxidation, laughing gas, chronic abuse, homocysteine, methylmalonic acid, holotranscobalamin

## Abstract

The recreational use of nitrous oxide (N_2_O) as “laughing gas” is a growing problem. The chronic toxicity of N_2_O is mainly due to its ability to oxidize vitamin B12, making it dysfunctional as a cofactor in metabolic pathways. This mechanism plays a major role in the development of neurological disorders in N_2_O users. The assessment of vitamin B12 status in N_2_O users is important but challenging due to the lack of decrease in total vitamin B12 in most cases despite genuine vitamin B12 functional deficiency. Other biomarkers, such as holotranscobalamin (holoTC), homocysteine (tHcy) and methylmalonic acid (MMA), are interesting candidates to properly assess vitamin B12 status. Here, we conducted a systematic review of case series in order to assess the prevalence of abnormal values of total vitamin B12, holoTC, tHcy and MMA in recreational N_2_O users, which is an important prerequisite for determining the best screening strategy in future guidelines. We included 23 case series (574 N_2_O users) from the PubMed database. Total circulating vitamin B12 concentration was low in 42.2% (95% confidence interval 37.8–46.6%, n = 486) of N_2_O users, while 28.6% (7.5–49.6%, n = 21) of N_2_O users had low circulating concentrations of holoTC. tHcy levels were elevated in 79.7% (75.9–83.5%, n = 429) of N_2_O users, while 79.6% (71.5–87.7%, n = 98) of N_2_O users had increased concentrations of MMA. In summary, the increases in tHcy and MMA were the most prevalent abnormalities, and should be measured alone or in combination in symptomatic N_2_O users rather than total vitamin B12 or holoTC.

## 1. Introduction

The recreational use of nitrous oxide (N_2_O) as “laughing gas” is a growing public health concern, especially in teenagers and young adults. For instance, 8.7% of 16-to-24-year-olds in the United Kingdom admitted to using N_2_O in 2019–20 [1]. In addition, three quarters of 600 health profession students in France have experienced N_2_O use [2]. N_2_O is recreationally consumed by inhalation for its rapid but short-lived feelings of euphoria and relaxation. The popularity of N_2_O consumption in teenagers and young adults is due to its easy availability, low price and the belief that it is a relatively safe and socially acceptable drug. 

Acute adverse effects following N_2_O inhalation are usually mild and transient. N_2_O use may produce nausea, vomiting, dizziness and tingling. Acute poisoning requiring medical treatment is relatively uncommon; medical treatment is generally required for injuries from falls caused by reduced motor coordination after inhalation. On the other hand, chronic use of N_2_O causes genuine dose-dependent toxicity. The chronic toxicity is mainly neurological, including symptoms of peripheral neuropathy, myelopathy and encephalopathy. The precise mechanisms of the neurological disorders are not fully understood, but alterations in vitamin B12 functions play an important role in N_2_O-induced demyelinating polyneuropathy. 

Vitamin B12 is an essential nutrient provided by foods of animal origin. It plays a major role in deoxyribonucleic acid synthesis and energy metabolism, and is therefore essential for hematopoiesis and nervous system functions such as myelin production. In actuality, vitamin B12 refers to four compounds called cobalamins, which are all characterized by a cobalt ion in the center of a corrin ring. The sixth coordination site with cobalt is variable, being either a cyano-, hydroxyl-, methyl- or 5’-deoxyadenosyl group, thus yielding cyano-, hydroxo-, methyl- and adenosyl- cobalamins, respectively. In the blood, approximately one-quarter of vitamin B12 binds to the specific transport protein transcobalamin to yield holotranscobalamin (holoTC), and the remaining vitamin B12 is carried by haptocorrin to form holohaptocorrin. HoloTC is able to transport vitamin B12 into the cells by binding to the transcobalamin receptor with high affinity and selectivity. On the contrary, holohaptocorrin does not bind the transcobalamin receptor due to electrostatic repulsion [3]. Thus, only holoTC is biologically active. As shown in Figure 1, methylcobalamin is a cofactor of methionine synthase (MTR), and a defect in the activity of methionine synthase increases levels of homocysteine. On the other hand, adenosylcobalamin acts as a cofactor for L-methylmalonyl-CoA mutase (MMCoAM), which converts methylmalonyl-CoA into succinyl-CoA. L-methylmalonyl-CoA mutase activity is impaired in vitamin B12 deficiency, which results in increased conversion of MMCoA into methylmalonic acid (MMA). 

N_2_O can be responsible for alterations in vitamin B12 functions due to its ability to inactivate it by oxidation, converting cobalt I (Co^+^) into cobalt III (Co^3+^) in the center of the corrin ring. The diagnosis of functional vitamin B12 deficiency in N_2_O users is challenging for routine clinical laboratories. The first-line biomarker to assess vitamin B12 status in clinical laboratories is serum vitamin B12, measured by immunoassays using intrinsic factor (the physiological ligand of vitamin B12 produced by stomach cells) to catch vitamin B12 from the whole serum or plasma. However, such immunoassays are not able to discriminate functional vitamin B12 from its oxidized form. Circulating levels of total homocysteine (tHcy) and MMA are other biomarkers of vitamin B12 deficiency, which appear relevant for the detection of dysfunctional vitamin B12. In addition, holoTC is sometimes considered more sensitive than total vitamin B12 for screening vitamin B12 deficiency [4]. It is unclear whether holoTC is useful in the diagnosis of functional vitamin B12 deficiency in N_2_O users. 

It is important to define the most effective strategy for properly screening N_2_O users with clinical signs for vitamin B12 deficiency. To know the prevalence of abnormal levels of vitamin B12 biomarkers in N_2_O users with clinical signs of toxicity is an important prerequisite for determining the best screening strategy in future guidelines. Here, we aimed to conduct a systematic review of case series to assess the prevalence of abnormal levels of total vitamin B12, tHcy, MMA and holoTC.

## 2. Materials and Methods

The systematic review was registered in the PROSPERO database (#CRD42023412939), and is presented according to the Preferred Reporting Items for Systematic Reviews and Meta-Analyses (PRISMA) 2020 guidelines [5].

### 2.1. Eligibility Criteria and Data Items

Table 1 shows the inclusion and exclusion criteria in the Population, Intervention, Comparison, Outcome, Study (PICOS) question format. We excluded records of 1 or 2 isolated cases because the risk of reporting an extreme phenotype is higher in isolated case reports than in case series. This may therefore lead to a significant bias of selection in the assessment of laboratory findings.

### 2.2. Information Sources and Search Strategy

The search procedure was conducted using the PubMed database. The search strategy was: (nitrous oxide[Title/Abstract]) AND ((vitamin B12[Title/Abstract]) OR (cobalamin[Title/Abstract]) OR (homocysteine[Title/Abstract]) OR (methylmalonic[Title/Abstract]) OR (methylmalonate[Title/Abstract])). We identified all reports from 1 January 2010 to 28 February 2023.

### 2.3. Selection and Data Collection Process

The two authors independently screened the titles and abstracts and discarded reports and studies that were not applicable. Relevant studies were selected regarding the inclusion and exclusion criteria. The two authors resolved disagreements on the eligibility of included studies through discussion and consensus. Data were independently collected in Excel sheets by the two authors for each predefined outcome. No automation tool was used in the process.

### 2.4. Study Risk of Bias Assessment

The risk of bias in the included studies was independently assessed by the two authors using the approach developed by Murad et al. in order to evaluate the methodological quality of the case series [6]. Discrepancies in judgement between the two authors were resolved through discussion. The risk of bias in the included studies is presented in Table 2. 

### 2.5. Effect Measures

The outcomes were calculated as the percentage of N_2_O users with low values of total vitamin B12 or holoTC and with increased values of tHcy or MMA. The 95% confidence intervals (CI) were calculated using the Wilson/Brown method. Proportions were compared using the Chi-square test. Statistical analyses and graphs were performed using GraphPad Prism 9.5.0.

## 3. Results

### 3.1. Study Selection

The PRISMA flowchart is shown in Figure 2. One hundred and eighty-nine reports were identified using our systematic approach, and twenty-three studies were ultimately included in the systematic review. The 23 selected studies included 574 N_2_O users. 

Seventy-three records were excluded because they did not report case series but only isolated cases. In addition, some studies were excluded since they seem to have included at least in part the same patients. Thus, we excluded the report of four cases by Joncquel Chevalier-Curt et al. [30], because it is highly likely that the participants were also included in another study from the same group including 52 N_2_O users [8]. In addition, five reports were published by the same team from the department of Neurology at Shenyang (China) without clear mention of overlapping [12,26,31,32,33]. Among these five records, we excluded the three reports from Fang et al. [31], Gao et al. [32] and Zheng et al. [33] due to similarities with the report from Yu et al. [12] in the demographic, clinical and biological characteristics of patients and also in the period of recruitment. Finally, we also excluded the case series from Temple and Horowitz because the N_2_O users took vitamin B12 supplements prior to blood sampling [34].

Among the 23 included studies, 21 (91.3%) were published in the last 5 years. The main characteristics of the selected studies are reported in Table 3. The studies originated from Europe (n = 9), Asia (n = 12) and Australia (n = 2). The systematic review included 574 N_2_O users (59% male). Nearly all users presented at care facilities due to neurological disorders.

### 3.2. Prevalence of Laboratory Abnormalities

The prevalence of laboratory abnormalities is reported in Figure 3. Firstly, we assessed the proportion of abnormal values of isolated biomarkers (Table 4). In the included case series, 42.2% (95% CI 37.8–46.6%) of the 486 N_2_O users had circulating concentrations of total vitamin B12 lower than the reference intervals. Elevated levels of tHcy were reported in 79.7% (95% CI 75.9–83.5%) of the 429 N_2_O users. MMA levels in blood or urinary samples were increased in 79.6% (95% CI 71.5–87.7%) of the 98 N_2_O users. The percentage of elevated tHcy and MMA in N_2_O users was statistically similar (*p* = 0.98). Values of holoTC concentrations were reported in only 21 individuals, and low levels were found in 28.6% (95% CI 7.5–49.6%) of them.

We also assessed the prevalence of abnormal values of combined biomarkers (Table 5). Fifteen studies provided data on the combination of biomarkers. High levels of both tHcy and MMA were found in 84.9% (95% CI 77.2–92.6%) of the 86 N_2_O users. This percentage did not significantly differ from the 79.7% of users with elevated tHcy (*p* = 0.27) or the 79.6% of users with high levels of MMA (*p* = 0.35). Thirty percent (95% CI 20.9–39.1%) of the 100 N_2_O users had abnormal levels of both total vitamin B12 and tHcy. 

## 4. Discussion

Low levels of total vitamin B12 are uncommon in the general population, especially among young adults such as N_2_O users. Data from the US NHANES 2003–2006 study showed that low vitamin B12 concentration affects between 7.7% and 19.4% of 19–39 year olds [35]. Here, we found that decreased levels of total vitamin B12 affected about 40% of symptomatic users of N_2_O in our systematic review of case series enrolling more than 500 users. Our result is fairly close to that of Marsden et al., who conducted a review of isolated case reports [36]. They found low levels of vitamin B12 in 56% of the 84 subjects [36]. A meta-analysis including 100 isolated case reports found a higher proportion of decreased levels of total vitamin B12, i.e., 71% [37]. Conversely, Garakani et al. observed low concentrations of vitamin B12 in only 20% in their review of 61 case reports [38]. The number of enrolled individuals and the selection of case series or case reports should explain the heterogeneity of results. 

In the present systematic review, we report normal levels of holoTC in more than two-thirds of the 21 N_2_O users. The available data were mainly provided by the study of Swart et al. [17]. However, the population enrolled in their study was representative of symptomatic users of N_2_O, and they reported a similar prevalence of abnormal values of total vitamin B12 and tHcy to us in our systematic review [17].

We mainly reported that levels of tHcy and MMA were increased in about 80% of N_2_O users in the present systematic review of case series. In their systematic review of case reports, Marsden et al. found elevated tHcy in 92% of the 53 users [36]. The meta-analysis by Oussalah et al., which included 100 isolated case reports, found a similar proportion of 90% [37]. In addition, Garakani et al. observed elevated concentrations of tHcy in 85% of users included from 20 case reports [36]. Regarding MMA levels, Marsden et al. reported high levels of MMA in 92% of the 38 users [36], while Oussalah et al. observed a proportion of 94% [37]. Garakani et al. found elevated concentrations of MMA in 83% of the 18 users selected from isolated case reports [36]. Our approach using case series rather than case reports likely explains the slightly lower value we observed here, due to the tendency to report more extreme phenotypes in case reports in compared to case series. 

The proportion of abnormal concentrations of tHcy and MMA are close in our systematic review, and the prevalence of elevated levels of both tHcy and MMA was not significantly higher than isolated parameters. On the other hand, measurement of tHcy is more common than measurement of MMA in clinical laboratories. This could lead to the conclusion that MMA is not as useful. However, cautions should be taken since elevated levels of tHcy could be also due to folate and vitamin B6 deficiencies or genetic defects. Folate is crucial for the remethylation of Hcy to methionine, while vitamin B6 acts as a cofactor of cystathione-β-synthase, involved in the transsulfuration of Hcy to cystathionine.

One of the strengths of our systematic review was the focus on cases series rather than case reports. Indeed, it is generally accepted that data obtained from case series are of the highest quality because there is a higher risk of selection bias in isolated case reports. In addition, to the best of our knowledge, our systematic review enrolled the largest number of N_2_O users included in case series or case reports reporting data on laboratory findings. Our systematic review has some limitations. Firstly, we could not account for vitamin B12, B6 and folate intakes in food, or for smoking or alcohol. These factors affect vitamin B12 status, and, therefore, probably modulate the prevalence of laboratory abnormalities in N_2_O users we reported here. Specific studies in subgroups could be interesting to further investigate the importance of these confounding factors. Secondly, we did not assess the influence of medications known to interact with vitamin B12 metabolism (e.g., proton pump inhibitors, colchicine and metformin). However, N_2_O users included in the selected studies were young, and drug interaction is unlikely to affect our findings. Finally, our results were not weighted by the risk of bias of the selected studies. However, we found a high risk of bias only for the study by Sluyts et al. regarding the selection of a very limited number of N_2_O users (n = 8) [13]. We therefore considered that our unweighted approach is appropriate for estimating the prevalence of laboratory abnormalities.

## 5. Conclusions

The medical community should be aware of how difficult it can be to interpret B12 status in the specific population of N_2_O users. The recreational use of N_2_O can mask a genuine vitamin B12 deficiency if we rely only on the circulating levels of B12 or holoTC. The results of our systematic review led us to conclude that a proper evaluation of vitamin B12 status in patients with a suspicion of N_2_O-induced toxicity should include at least tHcy. However, MMA, which performs as well as tHcy, is a better biomarker if the patient has another cause of tHcy elevation, such as folate deficiency. Regardless, precautions in the interpretation of tHcy and MMA should be taken since elevated levels of these biomarkers are also observed in kidney diseases. Thus, estimating glomerular filtration rate is a prerequisite for properly assessing tHcy and MMA in N_2_O users.

## Figures and Tables

**Figure 1 antioxidants-12-01191-f001:**
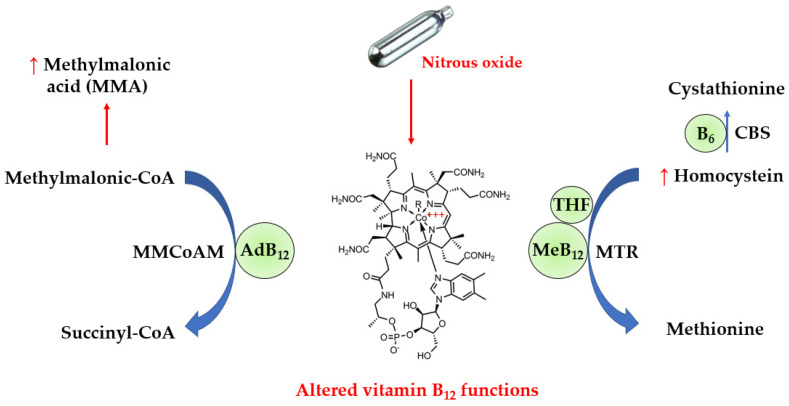
Alterations in biochemical pathways induced by N_2_O. N_2_O inactivates vitamin B12 via oxidation of the cobalt atom, leading to functional defects in vitamin B12 as a cofactor for the catabolism of homocysteine and methylmalonic acid. Thus, the circulating levels of methylmalonic acid and homocysteine can be increased in N_2_O-induced vitamin B12 deficiency. AdB_12_, adenosylcobalamin; B6, vitamin B6; CBS, cystathionine beta-synthase; MeB_12_, methylcobalamin; THF, 5-methyl tetrahydrofolate.

**Figure 2 antioxidants-12-01191-f002:**
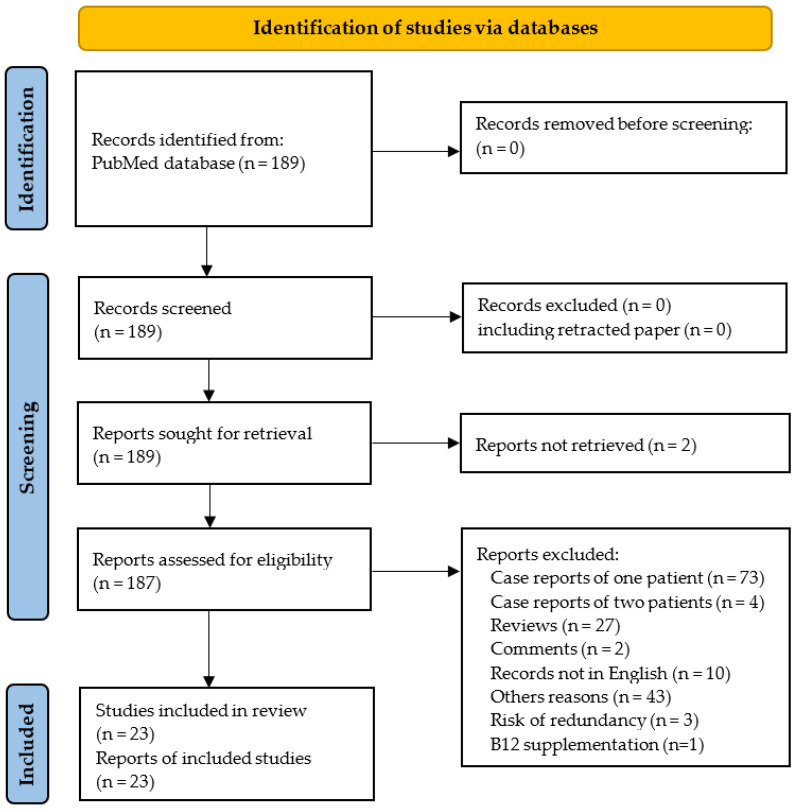
PRISMA 2020 flow diagram.

**Figure 3 antioxidants-12-01191-f003:**
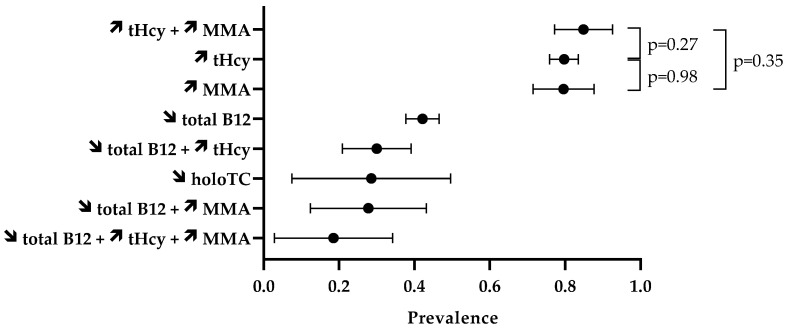
Prevalence of disturbed biomarkers. The symbols ↗ and ↘ mean increase and decrease, respectively. Data are means with 95% CI.

**Table 1 antioxidants-12-01191-t001:** Eligibility criteria for the selection of studies.

PICOS Parameter	Inclusion Criteria	Exclusion Criteria
Population	N_2_O users for recreational purposes with clinical symptoms attributable to N_2_O-related toxicity. Any age, gender or ethnicity.	N_2_O use in the context of anesthesia or other medical situations.
Intervention	Blood testing including at least one of the following biological parameters: total vitamin B12, tHcy, MMA or holoTC.	Vitamin B12 supplementation before blood collection.
Comparison	Comparison group not required.	
Outcomes	The primary outcomes were the proportion of N_2_O users with abnormal values of: Total vitamin B12 in serum/plasma; tHcy in plasma; MMA in plasma or urine; HoloTC in serum/plasma Abnormal values for each parameter were those defined in each study because cut-off limits are method-dependent.	Reports were excluded if the expression of results did not allow us to calculate the percentage of abnormal values.
Study design	Case series including at least 3 N_2_O users. Articles published in peer-reviewed journals in English.	Articles not published in English, case reports of one or two cases, meta-analyses, reviews, expert opinions, conference reports and studies in animal models.

**Table 2 antioxidants-12-01191-t002:** Assessment of methodological quality of the selected case series. Green color means low risk of bias, while orange color indicates that some concerns were noted in the study. Lastly, red color shows a high risk of bias.

Study	Selection	Ascertainment	Causality	Reporting	Overall Judgement
[7]					
[8]					
[9]					
[10]					
[11]					
[12]					
[13]					
[14]					
[15]					
[16]					
[17]					
[18]					
[19]					
[20]					
[21]					
[22]					
[23]					
[24]					
[25]					
[26]					
[27]					
[28]					
[29]					

**Table 3 antioxidants-12-01191-t003:** Characteristics of the selected studies.

Study	Location ^1^	Time	Recruited N_2_O Users	Age (y)	Gender (F/M)
[7]	Amsterdam, Netherlands (monocentric)	January 2015 to May 2021	17 users with thrombotic events	26 [range 18–53]	5/12
[8]	Lille, France (monocentric)	March 2020 to March 2022	52 users admitted to hospital	22	14/38
[9]	Linhai, China	May 2020 to June 2020	6 users	22 ± 4	2/4
[10]	Xi’an, China	May 2020 to November 2020	15 users with peripheral neuropathy	22 ± 5	8/7
[11]	Lille, France (multicentric)	January 2019 to August 2020	20 users with neuropathy	19 [range 16–34]	17/3
[12]	Shenyang, China (monocentric)	January 2018 to December 2020	110 users with neuropathy	21 ± 4	53/57
[13]	Edegem, Belgium (monocentric)	N.P.	8 users with neuropathy in limbs	22 ± 4	2/6
[14]	Xuzhou, China (monocentric)	January 2017 to December 2020	61 users with neuropathy	22 ± 3	19/42
[15]	Shenyang, China (monocentric)	February 2017 to July 2020	63 users with neuropathy	23 ± 4	25/38
[16]	Paris, France	July 2020 to April 2021	7 users referred for electroneuromyography	21 ± 4	1/6
[17]	Sydney, Australia (multicentric)	2016 to 2020	20 users with myeloneuropathy	24 (range 18–40)	11/9
[18]	Bobigny, France (monocentric)	August 2020 to April 2021	12 users with spinal cord injury and/or peripheral neuropathies	22 ± 3	6/6
[19]	Strasbourg, France (monocentric)	April 2020 to February 2021	5 users with neuropathy	24 ± 4	2/3
[20]	Hefei, China (multicentric)	October 2018 to May 2020	20 users with neuropathy	23 (IQR 20–28)	9/11
[21]	London, UK (monocentric)	N.P.	3 users with peripheral neuropathy	21 ± 2	2/1
[22]	Qingdao, China (monocentric)	January 2016 to August 2019	21 users with neuropathy	22 ± 5	7/14
[23]	Hanoi, Vietnam	May 2018 to July 2019	47 users admitted to hospital	24 ± 6	24/23
[24]	Xuzhou, China	2015 to 2019	33 users with neuropathy	22 ± 3	4/29
[25]	Melbourne, Australia (monocentric)	N.P.	4 users with neuropathy	20 ± 3	4/0
[26]	Shenyang, China (monocentric)	January 2014 to June 2019	4 users with neuropathy and skin hyperpigmentation	20 ± 3	3/1
[27]	London, UK (monocentric)	November 2016 to May 2017	10 users with symptoms of subacute degeneration of the spinal cord	22 (range 17–26)	3/7
[28]	Taoyuan, Taiwan (monocentric)	2005 to 2015	33 users with myeloneuropathy	23 ± 3	14/19
[29]	Taiwan	N.P.	3 users with myeloneuropathy and peripheral neuropathy	21 ± 3	2/1

IQR, inter-quartile range. ^1^ The monocentric or multicentric design of selected studies was presented in the table only if the information was unambiguous in the report.

**Table 4 antioxidants-12-01191-t004:** Prevalence of disturbed biomarkers in the selected studies. Data are reported as percentage of N_2_O users with abnormal levels of biomarker (number of N_2_O users with abnormal level/total number of N_2_O users with available laboratory data). The symbols ↗ and ↘ mean increase and decrease, respectively.

Study	Participants	↘ Total B12	↗ tHcy	↗ MMA	↘ HoloTC
[7]	17	46% (6/13)	89% (8/9)	N.P.	N.P.
[8]	52	56% (29/52)	98% (51/52)	75% (39/52)	N.P.
[9]	6	67% (4/6)	50% (3/6)	N.P.	N.P.
[10]	15	33% (3/9)	N.C.	N.P.	N.P.
[11]	20	64% (9/14)	100% (13/13)	100% (7/7)	N.P.
[12]	110	60% (34/57)	69% (31/45)	N.P.	N.P.
[13]	8	13% (1/8)	100% (8/8)	U: 88% (7/8)	0% (0/4)
[14]	61	44% (20/45)	68% (27/40)	N.P.	N.P.
[15]	63	35% (22/63)	87% (55/63)	N.P.	N.P.
[16]	7	14% (1/7)	100% (6/6)	N.P.	N.P.
[17]	20	50% (10/20)	83% (10/12)	N.P.	35% (6/17)
[18]	12	33% (4/12)	100% (11/11)	100% (11/11)	N.P.
[19]	5	0% (0/5)	100% (5/5)	100% (4/4)	N.P.
[20]	20	25% (5/20)	70% (14/20)	N.P.	N.P.
[21]	3	66% (2/3)	100% (2/2)	100% (1/1)	N.P.
[22]	21	17% (3/18)	78% (14/18)	U: 29% (2/7)	N.P.
[23]	47	57% (27/47)	87% (41/47)	N.P.	N.P.
[24]	33	27% (9/33)	82% (27/33)	N.P.	N.P.
[25]	4	100% (4/4)	100% (1/1)	N.P.	N.P.
[26]	4	100% (4/4)	100% (4/4)	N.P.	N.P.
[27]	10	40% (4/10)	N.P.	88% (7/8)	N.P.
[28]	33	9% (3/33)	30% (10/33)	N.P.	N.P.
[29]	3	33% (1/3)	100% (1/1)	N.P.	N.P.
**Total**	**574**	**42.2% (205/486)**	**79.7% (342/429)**	**79.6% (78/98)**	**28.6% (6/21)**

N.P., not provided; U, urine.

**Table 5 antioxidants-12-01191-t005:** Prevalence of disturbed combined biomarkers. Data are reported as percentage of N_2_O users with abnormal levels of biomarker (number of N_2_O users with abnormal level/total number of N_2_O users with available laboratory data). The symbols ↗ and ↘ mean increase and decrease, respectively.

Study	Participants	↘ Total B12 + ↗ tHcy	↘ Total B12 + ↗ MMA	↗ tHcy + ↗ MMA	↘ Total B12 + ↗ tHcy + ↗ MMA
[7]	17	33% (3/9)	N.P.	N.P.	N.P.
[8]	52	N.P.	N.P.	75% (39/52)	N.P.
[9]	6	50% (3/6)	N.P.	N.P.	N.P.
[11]	20	N.P.	N.P.	100% (7/7)	N.P.
[34]	4	0% (0/4)	0% (0/4)	100% (4/4)	0% (0/4)
[13]	8	13% (1/8)	13% (1/8)	U: 88% (7/8)	13% (1/8)
[16]	7	14% (1/7)	N.P.	N.P.	N.P.
[18]	12	36% (4/11)	36% (4/11)	100% (11/11)	36% (4/11)
[19]	5	0% (0/5)	0% (0/4)	100% (4/4)	0% (0/4)
[20]	20	40% (4/10)	N.P.	N.P.	N.P.
[21]	3	50% (1/2)	100% (1/1)	100% (1/1)	N.P.
[25]	4	100% (1/1)	N.P.	N.P.	N.P.
[26]	4	100% (4/4)	N.P.	N.P.	N.P.
[27]	10	N.P.	50% (4/8)	N.P.	N.P.
[28]	33	24% (8/33)	N.P.	N.P.	N.P.
**Total**	**205**	**30% (30/100)**	**28% (10/36)**	**85% (73/86)**	**19% (5/27)**

N.P., not provided.

## Data Availability

The data presented in this study are available within the article.

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
