# Peer review of "Vitamin B12 Status in Recreational Users of Nitrous Oxide: A Systematic Review Focusing on the Prevalence of Laboratory Abnormalities"

_antioxidants, 2023, doi:10.3390/antiox12061191_

Round 1
Reviewer 1 Report
This is a helpful compilation review of published case studies on patients with neurological abnormalities who are recreational users of N2O. The manuscript is well written but there are some sentences where the meaning is difficult to immediately determine. These are particularly sentences where there is the grammatical use of the idiom: “… of X and Y respectively”. These difficulties are listed below:
Lines19-21: “44.8% (95% confidence interval 40.5-49.1%, n=511 N2O users) and 28.6% (7.5-49.6%, n=21) of N2O users had low circulating concentrations of total vitamin B12 and holoTC, respectively.”
It would be more easily understood if the sentence were revised to something like the following: “Total circulating vitamin B12 concentration was low in 44.8% (40.5 – 49.1% 95% confidence interval, n=511) N2O users, while 28.6% (7.5 – 49.6%, n=21) of N2O users had low circulating concentrations of holoTC.”
Lines 21-22: A similar reconstruction of the sentence would help understanding by a reader.
Lines 123-124: A similar reconstruction of this list would make the meaning of each colour easier to understand.
Lines 78-80: “To know if holoTC is useful in the diagnosis of functional vitamin B12 deficiency in N2O users remains unclear.”
It would be easier to understand this sentence if written something like: “It is unclear whether holoTC is useful in the diagnosis of functional vitamin B12 deficiency in N20 users”
As explained in the comments to the authors, there are several sentences where the sentence structure tends to obscure the meaning.
Author Response
This is a helpful compilation review of published case studies on patients with neurological abnormalities who are recreational users of N2O. The manuscript is well written but there are some sentences where the meaning is difficult to immediately determine. These are particularly sentences where there is the grammatical use of the idiom: “… of X and Y respectively”. These difficulties are listed below:
Lines19-21: “44.8% (95% confidence interval 40.5-49.1%, n=511 N2O users) and 28.6% (7.5-49.6%, n=21) of N2O users had low circulating concentrations of total vitamin B12 and holoTC, respectively.”
It would be more easily understood if the sentence were revised to something like the following: “Total circulating vitamin B12 concentration was low in 44.8% (40.5 – 49.1% 95% confidence interval, n=511) N2O users, while 28.6% (7.5 – 49.6%, n=21) of N2O users had low circulating concentrations of holoTC.”
Lines 21-22: A similar reconstruction of the sentence would help understanding by a reader.
We thank the Reviewer for helping us to clarify the meaning of some sentences. We changed the two sentences in the revised abstract following the Reviewer’s proposal (lines 21-26). The values changed slightly due to comments from another Reviewer.
Lines 123-124: A similar reconstruction of this list would make the meaning of each colour easier to understand.
We also changed this sentence in the heading of Table 2 (lines 133-134).
Lines 78-80: “To know if holoTC is useful in the diagnosis of functional vitamin B12 deficiency in N2O users remains unclear.” It would be easier to understand this sentence if written something like: “It is unclear whether holoTC is useful in the diagnosis of functional vitamin B12 deficiency in N20 users”
We followed the Reviewer’s proposal in the revised manuscript (lines 87-88).
Comments on the Quality of English Language
As explained in the comments to the authors, there are several sentences where the sentence structure tends to obscure the meaning.
We improved several sentences in order to clarify the meaning (see above).
Reviewer 2 Report
I have now reviewed the manuscript above entitled: Vitamin B12 Status in Recreational Users of Nitrous Oxide: A Systematic Review Focusing on the Prevalence of Laboratory Abnormalities.
I think the topic is important and worth researching further. The authors have done their best to conduct the systematic review, though I was surprised that they have not done a meta-analysis.
I have the following concerns:
1. Where are the controls? Have the authors investigated some healthy individuals to compare the levels? It is worth mentioning to readers the reference ranges for all the parameters discussed.
2. Was there any confounding factors that might have interfered with the outcomes?
3. What are the limitation of the studies analyzed? and was there any effects of these on results obtained?
4. What are the effects of bias in Table 2 on the outcomes?
5. Prevalence of disturbed laboratory biomarkers (Fig2 and Tables 4 and 5) are confusing and require further clarifications. What are the effects of these on the overall results?
7. What were other drugs and medications taken by participants which might have interfered with lab biomarkers and stats?
8. The statement in discussion: "The prevalence of decreased levels of total vitamin B12 is quite low among symptomatic users of N2O, affecting less than half of individuals in our systematic review" Could you explain why it is low? Could this be rectified by estimating levels of homocysteine and methylmalonic acid.
Author Response
I have now reviewed the manuscript above entitled: Vitamin B12 Status in Recreational Users of Nitrous Oxide: A Systematic Review Focusing on the Prevalence of Laboratory Abnormalities.
I think the topic is important and worth researching further. The authors have done their best to conduct the systematic review, though I was surprised that they have not done a meta-analysis.
I have the following concerns:
We are grateful to the Reviewer for helping us to improve the manuscript.
- Where are the controls? Have the authors investigated some healthy individuals to compare the levels? It is worth mentioning to readers the reference ranges for all the parameters discussed.
Obviously, the case series of symptomatic N2O users we selected in our systematic review did not report data on healthy individuals. But, low levels of total vitamin B12 are uncommon in the general population, especially among young adults such as N2O users. Data from the US NHANES 2003-2006 study showed that low vitamin B12 concentration affects between 7.7% and 19.4% of 19 to 39 year olds (DOI: 10.3945/ajcn.2009.28401). We mentioned this prevalence as a reference in the Discussion section of the revised manuscript (lines 200-203).
The reference intervals of studied parameters are method dependent and, therefore, no universal cut-off limits can be stated. We therefore chose to systematically refer to the thresholds defined by the authors in each study. We pointed out this methodological concern in the Table 1 of the Materials and Methods section of the revised manuscript (line 105).
- Was there any confounding factors that might have interfered with the outcomes?
The intake of vitamins (especially B12, B9 and B6) in the food or supplements affect vitamin B12 status in the general population and also in N2O users. We excluded studies where N2O users were taking vitamin B12 supplementation prior to blood collection (see Table 1). Some medications interacting with vitamin B12 metabolism could also affect biomarkers of vitamin B12 status. All these variables could modulate the outcomes. We discussed these confounding factors as limitations of our review in the Discussion section (lines 242-249).
- What are the limitation of the studies analyzed? and was there any effects of these on results obtained?
As reported in Table 2, we found a high risk of bias only for the study by Sluyts et al. regarding the selection of N2O users. However, we considered unnecessary to weight our results since this study included a limited number of participants (n=8), leading to a very low influence on overall findings. We mentioned this concern as a limitation of our review in the Discussion section of the revised manuscript (lines 249-253).
- What are the effects of bias in Table 2 on the outcomes?
As mentioned just above, we considered that the bias of selection we reported for the study of Sluyts et al. has very low influence on overall findings. We discussed this point in the revised manuscript (lines 249-253).
- Prevalence of disturbed laboratory biomarkers (Fig2 and Tables 4 and 5) are confusing and require further clarifications. What are the effects of these on the overall results?
We provided further clarifications on the calculation of the prevalence of disturbed laboratory biomarkers in the Table 1 of M&M section (line 105). Moreover, we added explanation in the headings of the Tables 4 and 5 in order to help the readers to understand the tables.
- What were other drugs and medications taken by participants which might have interfered with lab biomarkers and stats?
Some medications can interfere with vitamin B12 metabolism. We excluded studies in which N2O users were taking vitamin B12 supplementation prior to blood collection. We were not able to quantitatively estimate the effect of drug interaction since few data are reported in the selected studies. However, we identified this concern as a limitation of our review at the end of the revised manuscript (lines 246-249).
- The statement in discussion: "The prevalence of decreased levels of total vitamin B12 is quite low among symptomatic users of N2O, affecting less than half of individuals in our systematic review" Could you explain why it is low? Could this be rectified by estimating levels of homocysteine and methylmalonic acid.
In accordance with the reply to the question #1 of the Reviewer, we found more appropriate to remove the word “low” in the revised manuscript (line 203). More than 40% of N2O users had low levels of total vitamin B12 in our review. Obviously, it is much higher than in general population, as discussed in the reply to the question #1 of the Reviewer.
Reviewer 3 Report
The short review by Tanguy Ménétrier und Damien Denimal entitled: “Vitamin B12 status in recreational users of nitrous oxide: a systematic review focusing on the prevalence of laboratory abnormalities” present the screening of Pubmed databank for papers that deal with the measurement of functional vitamin B12 in recreational N2O users. 189 papers were retrieved and only 23 of them were included in this review for matching several selection criteria presented in Table 1, most notably focusing on selecting case series rather than case reports. The aim was to get serious measurements of vitamin B12 (or cobalamin) or homocysteine and methylmalonate, two biomarkers of vitamin B12 deficiency, for setting in a clinical laboratory the best method to accurately assign the levels of functional vitamin B12 in a N2O users presenting detrimental symptoms. Clearly, measuring the blood levels of homocysteine and methylmalonate seems the best choice to evaluate clinically a deficiency in functional vitamin B12 caused by oxidation of the cobalt ion by N2O. Overall, I did enjoy reading this short review und found the paper is interesting and maybe publishable in antioxidants as such. I have however a few minor comments that I would like to be answered/commented by the authors.
Comments
1. There are two figures 1 in this manuscript: figure 1: one at page 2 of 11, and the other at page 5 of 11.
2. Some abbreviations are not defined and thus not easily understandable by the average reader, such as MeB12 at line 57, or B6 in figure 1. In the Tables, what does the abbreviation NP stand for?
3. Lines 73-74. What precisely is the “intrinsic factor” that, if I understood correctly this sentence, is a ligand of vitamin B12?
4. The fact that in the second figure 1 (PRISMA 2020 flow program), the “included” series present two different values: n=23 and n=24. What is the difference? How is it that the number of studies differ from unity with the number of reports? It is not sufficiently defined in the main text.
5. The total N2O users in the cohort gathered by the authors in this study is n = 596. The average reader, including this reviewer, will have some difficulties to clearly understand why this number falls to n = 511 in line 160, and also elsewhere in the main text. What do these discrepancies in the numbers cover? The authors should be more specific in the main text for explaining these points.
6. The Discussion section begins by a “Firstly” (line 185) that is not followed anywhere else by a “secondly”. Please discard it.
Question
At lines 52-56 of their review, the authors write that about ¼ of serum vitamin B12 is bound to transcobalamin and the remaining ¾ to haptocorrin. Only vitamin B12 bound to transcobalamin enters the cells. It is known that transcobalamin and haptocorrin have very similar two-domain 3D structures, highly superimposable (See: McCorvie TJ et al. The complex machinery of human cobalamin metabolism. J Inherit Metab Dis 2023, 1-5). Could the authors comment on how these two vit B12-binding proteins with so similar structures can differ so greatly in their behavior towards vitamin B12 intake into the cells?
Author Response
The short review by Tanguy Ménétrier and Damien Denimal entitled: “Vitamin B12 status in recreational users of nitrous oxide: a systematic review focusing on the prevalence of laboratory abnormalities” present the screening of Pubmed databank for papers that deal with the measurement of functional vitamin B12 in recreational N2O users. 189 papers were retrieved and only 23 of them were included in this review for matching several selection criteria presented in Table 1, most notably focusing on selecting case series rather than case reports. The aim was to get serious measurements of vitamin B12 (or cobalamin) or homocysteine and methylmalonate, two biomarkers of vitamin B12 deficiency, for setting in a clinical laboratory the best method to accurately assign the levels of functional vitamin B12 in a N2O users presenting detrimental symptoms. Clearly, measuring the blood levels of homocysteine and methylmalonate seems the best choice to evaluate clinically a deficiency in functional vitamin B12 caused by oxidation of the cobalt ion by N2O. Overall, I did enjoy reading this short review und found the paper is interesting and maybe publishable in antioxidants as such. I have however a few minor comments that I would like to be answered/commented by the authors.
Comments
We are grateful to the Reviewer for helping us to improve the manuscript.
- There are two figures 1 in this manuscript: figure 1: one at page 2 of 11, and the other at page 5 of 11.
We thank the Reviewer for identifying this error. We therefore corrected the numbering of the figures (and also in the body text).
- Some abbreviations are not defined and thus not easily understandable by the average reader, such as MeB12 at line 57, or B6 in figure 1. In the Tables, what does the abbreviation NP stand for?
We have replaced the abbreviations “MeB12” and “AdB12” with the entire words in the revised manuscript. We also clarified “B6” and “NP” in the legends of the Figure 1 and tables.
- Lines 73-74. What precisely is the “intrinsic factor” that, if I understood correctly this sentence, is a ligand of vitamin B12?
The intrinsic factor is the physiological ligand of vitamin B12, which is produced by stomach cells and is necessary for intestinal absorption of vitamin B12. The intrinsic factor is used in competitive immunoassays to catch vitamin B12 from the whole serum. We added this information in the revised manuscript (lines 81-82).
- The fact that in the second figure 1 (PRISMA 2020 flow program), the “included” series present two different values: n=23 and n=24. What is the difference? How is it that the number of studies differ from unity with the number of reports? It is not sufficiently defined in the main text.
We initially identified 5 reports based on the same cohort of N2O users by a team at Shenyang, China (lines 148-150). Among these 5 reports, we initially maintained the reports of Zheng et al. (ref #33 in the revised manuscript) and Yu et al. (ref #12) because they did not show the same laboratory outcomes. Therefore, we initially included two reports from one study.
But, by checking the concern mentioned by the Reviewer, we evidenced that we made a confusion between two reports from two different first authors, both called Zheng (ref #22 and #33). We corrected this error, and excluded the reference #33 in our review due to overlapping with Yu et al. (ref #12). Therefore, the number of reports and studies are now the same. In addition, the replacement of the study of Zheng et al. (ref #33) by the study of Zheng et al. (ref #22) induced a novel calculation of some stats (lines 170-172 and Fig. 3).
- The total N2O users in the cohort gathered by the authors in this study is n = 596. The average reader, including this reviewer, will have some difficulties to clearly understand why this number falls to n = 511 in line 160, and also elsewhere in the main text. What do these discrepancies in the numbers cover? The authors should be more specific in the main text for explaining these points.
We reported 596 participants, corresponding to all N2O users recruited in the selected studies. However, the levels of total vitamin B12 were available only for 511 participants of them. We improved the meaning in the heading of the Tables 4 and 5. We also mentioned in the body text that 596 N2O users correspond to the number of patients enrolled in the 23 selected studies (line 139).
- The Discussion section begins by a “Firstly” (line 185) that is not followed anywhere else by a “secondly”. Please discard it.
We discarded the word “firstly” (line 203).
Question
At lines 52-56 of their review, the authors write that about ¼ of serum vitamin B12 is bound to transcobalamin and the remaining ¾ to haptocorrin. Only vitamin B12 bound to transcobalamin enters the cells. It is known that transcobalamin and haptocorrin have very similar two-domain 3D structures, highly superimposable (See: McCorvie TJ et al. The complex machinery of human cobalamin metabolism. J Inherit Metab Dis 2023, 1-5). Could the authors comment on how these two vit B12-binding proteins with so similar structures can differ so greatly in their behavior towards vitamin B12 intake into the cells?
The cellular uptake of holotranscobalamin is mediated by the plasma membrane transcobalamin receptor (TCblR) encoded by the CD320 gene. TCblR/CD320 has been reported to bind holotranscobalamin with high affinity but not haptocorrin although these two molecules have strongly overlapping 3D structures, as mentioned by the Reviewer. It has been shown that the different electrostatic potentials at the relevant surfaces of transcobalamin and haptocorrin likely explain the difference in their ability to bind to the TCblR/CD320 receptor (Alam et al. Nat Commun 2016, doi: 10.1038/ncomms12100). In other words, haptocorrin cannot bind CD320 due to electrostatic repulsion. We commented this point in the introduction section of the revised manuscript, and mentioned Alam’s paper (lines 60-62).
Reviewer 4 Report
The manuscript submitted to antioxidants is an interesting and thorough review about vitamin B12 status and use of N2O. This is a very well crafted work and the reviewer would merely like to propose a few points of consideration by the authors with the intention to improve the manuscript.
1. One point of consideration could be the nutritional status of the participants and diet/nutrition in general. When we are considering vitamin B12 status it is important to consider the dietary intake of vitamin and/or supplementation as well as potential smoking and alcohol intake. These can all function as potential confounding factors which can in turn alter the perceived status of vitamin B12.
2. While the authors do make a mention of that it would strengthen the paper to illustrate more clearly the importance of folate and other vitamin/mineral status as relating to vitamin B12.
Good job overall!
English language is OK in the narrative.
Author Response
The manuscript submitted to antioxidants is an interesting and thorough review about vitamin B12 status and use of N2O. This is a very well crafted work and the reviewer would merely like to propose a few points of consideration by the authors with the intention to improve the manuscript.
- One point of consideration could be the nutritional status of the participants and diet/nutrition in general. When we are considering vitamin B12 status it is important to consider the dietary intake of vitamin and/or supplementation as well as potential smoking and alcohol intake. These can all function as potential confounding factors which can in turn alter the perceived status of vitamin B12.
We fully agree with the Reviewer that the intake of vitamins (especially B12, B9 and B6) in the food or supplements affect vitamin B12 status in N2O users. It is also the case for tabacco and alcohol, as mentioned by the Reviewer. Thus, these variables are potential confounding factors in estimating the prevalence of laboratory abnormalities. For this reason, we excluded studies in which N2O users were taking vitamin B12 supplementation prior to blood collection. We were not able to quantitatively estimate the effect of these factors since none of the selected studies reported the amount of vitamin B12 in the diet, and very few data were available on tabacco and alcohol consumption in the selected case series. However, we have identified these potential confounding factors as limitations to the results of our systematic review in the revised manuscript (lines 242-246).
- While the authors do make a mention of that it would strengthen the paper to illustrate more clearly the importance of folate and other vitamin/mineral status as relating to vitamin B12.
Both folate and vitamin B6 play a significant role relating to markers of vitamin B12 status. Especially, folate and vitamin B6 play a role for homocysteine catabolism. In aprticular, vitamin B6 acts as cofactor of cystathione synthase, while folate is crucial for methionine synthase in the remethylation pathway. Thus, folate or vitamin B6 deficiencies could lead to elevated levels of tHcy, independently to vitamin B12 deficiency. We strengthened the importance of folate and vitamin B6 status in assessing vitamin B12 status in the revised manuscript (lines 234-236 and lines 243).
Good job overall!
Round 2
Reviewer 2 Report
I have now reviewed all the responses of authors to my comments on the above manuscript, Antioxidants (ISSN 2076-3921).
I am satisfied that the authors have done their best to address all the issues of concern. Therefore I can now accept the manuscript in its present revised form.